# Prospects for Bioenergy Development Potential from Dedicated Energy Crops in Ecuador: An Agroecological Zoning Study

**Christian R. Parra [1], Angel D. Ramirez [2,*], Luis Manuel Navas-Gracia [1,*], David Gonzales [2] and Adriana Correa-Guimaraes [1]**

1   TADRUS Research Group, Department of Agricultural and Forestry Engineering, University of Valladolid (UVa), Campus Universitario de Palencia, Avenida de Madrid, 50, 34004 Palencia, Spain
2   Escuela Superior Politecnica del Litoral, ESPOL, Facultad de Ingeniería en Mecánica y Ciencias de la Producción, Campus Gustavo Galindo, Km. 30.5 Vía Perimetral, P.O. Box 09-01-5863, Guayaquil EC090902, Ecuador
*   Correspondence: aramire@espol.edu.ec (A.D.R.); luismanuel.navas@uva.es (L.M.N.-G.)

**Abstract:** Most climate change mitigation scenarios rely on the incremental use of biomass as energy feedstock. Therefore, increasing the share of alternative sustainable energy sources as biomass is crucial to provide both peak and base electricity loads in future scenarios. The bioenergy potential of Ecuador has been addressed for agricultural by-products but not for dedicated bioenergy crops. Agricultural zoning studies have been developed for food crops but not for energy crops. Currently, the bioenergy share of electricity produced in Ecuador (1.4%) comes mainly from the use of sugar cane bagasse from sugar production. This study aims to identify potential sustainable bioenergy resources for continental Ecuador using agroecological zoning methodologies and considerations regarding land management, food security, in-direct land use change and ecological and climate change risks. The results identified 222,060.71 ha available to grow dedicated bioenergy crops and potential electricity production of 8603 GWh/year; giant reed ranks first with a potential net energy yield of 4024 GWh per year, and Manabí province presents the highest potential with 3768 GWh/year. Large-scale deployment of bioenergy in Ecuador would require the study of sustainability considerations of each project. The species studied are traditional bioenergy crops; research on novel species is encouraged.

**Keywords:** electricity; bioenergy; biomass; land; climate change; availability; sustainability; GIS; Ecuador; Latin America



## 1. Introduction

Climate change is one of the most serious threats to sustainability [1]. All of the climate change mitigation scenarios developed by the Intergovernmental Panel on Climate Change (IPCC) report that energy sources and climate change mitigation rely on large-scale bioenergy deployment [2]. Bioenergy could supply 15% of global primary energy by 2050 [3].Worldwide, 1.4 billion hectares have been identified as suitable to be allocated for bioenergy production. In Latin America, there are currently 343 surplus million hectares, which could be dedicated to this end [4].

Under the current greenhouse gas (GHG) accounting systems, emissions produced when biomass is burnt for energy are accounted as zero, resulting in what is referred to as the carbon neutrality assumption [5]. Life cycle assessment (LCA) studies show a lower carbon footprint for most bioenergy feedstocks when compared with reference fossil fuel alternatives [6,7]. Bioenergy is considered a key component for the transition from a petroleum-based economy [6] and complementary to the circular economy concept [7,8].

Nowadays, the increase in demand for bioenergy production has led to various environmental and socio-economic concerns. It has been linked to the food crisis and sparked the food versus fuel debate [9]. In addition, bioenergy with carbon capture and

storage (BECCS), the combination of carbon capture and sequestration (CCS), and the use of biomass as an energy source to mitigate $CO_2$ emissions has been identified as a negative GHG emission technology (NET) [10–12]. World energy prospective models under climate change mitigation scenarios indicate that the large-scale deployment of biomass (>200 EJ), together with BECCS, could help to keep global warming 2° degrees below pre-industrial levels [13].

According to the nationally determined contribution (NDC) of Ecuador, the country could reduce its aggregated greenhouse emissions from energy, agriculture, industrial processes, and waste sectors by the year 2025 compared with the 2008 levels by 9% on an unconditional scenario and 20.9% in a conditional scenario supported by international cooperation [14].

In Ecuador, the electricity demand will increase from 65 to 74 TWh/year by 2050 [15]. Under all of the climate change scenarios, an expansion of hydropower capacity must be complemented by other baseload generation technologies such as natural gas or renewables such as biomass and geothermal energy to provide both peak and baseload generation. The uncertainty in hydropower generation implies an increase in terms of cumulative GHG emissions due to the participation of fossil-fuel-derived energy which could increase the global warming potential of net electricity generation in Ecuador from 12 to 20 times by 2050 over the 2016 level [16]. In the country, most of the fossil-based electricity production comes from fuel oil steam power (FO-SP) technology which presents the highest contribution in terms of greenhouse gas emissions [17]. In addition, to achieve deep decarbonization in the country, it is necessary to foster strategic climate change mitigation scenarios in which bioenergy and reforestation play a main role. To achieve its climate change mitigation goals, Ecuador's energy matrix must be diversified with higher shares of low carbon technologies and electrification of energy end-use in the transport, buildings, and industry sectors [18]. Therefore, the participation of bioenergy in the future energy mix of Ecuador could alleviate possible constraints regarding hydropower deployment and reduce fossil fuels' participation in electricity generation.

In 2020, biomass-based electricity accounted for 1.4% of the Ecuadorian electricity mix. The main producers of electricity derived from biomass are the sugar mills which sell electricity derived from sugar cane bagasse combustion to the Ecuadorian electricity system [19]. The total installed capacity of electricity generation is 136.4 MW [20]. There are some experiences of energy recovery using landfill gas in Ecuador: Pichacay in Cuenca and Chachoan in Ambato. In Quito city, the El Inga landfill site burns methane, but no energy is produced [21]. There is no legal framework to promote the exploitation of solid wastes for the development of waste-to-energy power plants. Nevertheless, some research has been performed [22]. Regarding liquid biofuels, during 2010, the Ecopaís program was launched to start the distribution of a 10% ethanol/gasoline blend in Guayaquil city. Since the beginning of the pilot project, 52,771.025 gallons of anhydrous ethanol have been produced [23]. The environmental impact of some biofuels and bioelectricity systems in Ecuador has been studied with a life cycle [24–27].

Furthermore, the Bioenergy Atlas of Ecuador, developed in 2015, addressed the energy generation potential of the main agricultural residues [28]. A total gross energy production of 199,057.23 TJ/year was estimated. The highest residue-based bioenergy potential comes from the following commodities: palm oil, banana, rice, cocoa, corn, and plantain. On the other hand, the energy potential from livestock waste was estimated to be 73.78 TJ/year [28].

Geographical information systems (GIS) [29] have been used for a wide range of zoning applications, e.g., the estimation of agroclimatic conditions' influence on the productivity of promising bioenergy crops [30] and the production possibilities of specific types of plants to punctual soil characteristics [31]. Other authors have used GIS to evaluate bioenergy production zones based on their energy efficiency using indicators such as energy return for energy invested (EROEI) [32].

Some authors have used GIS applications for estimating the bioenergy potential of agricultural residues [33] and municipal solid waste [34] and to assess the geographic distribution of residual forest biomass with energy ends [35].

More complex zoning models have integrated GIS applications and different approaches such as life cycle assessment (LCA) for the sustainable bioenergy planning of agriculture residues, taking into account its environmental impact potential [36] and dynamic yield simulation models; for determining regional bioenergy potentials [37], fuzzy logic, network optimization, and dynamic yield simulation models; and for determining regional bioenergy potentials [38].

Moreover, the future long-term impacts on water, erosion, sedimentation, and agricultural chemicals release due to crop management have also been studied within bioenergy zoning. Studies [39,40] have used the soil and water assessment tool (SWAT) developed by the U.S. Department of Agriculture to address these impacts [41].

In addition, the potential for producing bioenergy worldwide in marginal lands has been addressed in several studies [42–46]. Zoning studies for bioenergy resources in Latin America have been performed for agricultural by-products [47,48] and second-generation energy crops [30,49,50].

Finally, agroecological zoning studies for Ecuador have been developed mainly for food commodities [51,52].

Nevertheless, to the authors' knowledge, no study to date has identified the bioenergy potential of biomass-production-dedicated crop alternatives to be produced in Ecuador for electricity generation.

It must be mentioned that zoning studies for variable renewable energy resources as solar [53], and wind [54,55].

Considering the role of bioenergy in decarbonization scenarios in the country, according to [18], it is essential to explore the development potential of dedicated bioenergy feedstocks to secure a low-carbon baseload. Nevertheless, as prescribed by [9,56], bioenergy development should not compete with food production and natural ecosystems. Therefore, the main aim of this study is to quantify the energy potential of dedicated bioenergy crops in Ecuador. The specific objectives are (i) to identify dedicated bioenergy crops that could be produced in Ecuador, (ii) to identify available land with suitable agroecological conditions to produce dedicated bioenergy crops within a non-competitive scheme with food and natural ecosystems, and (ii) to quantify the gross bioenergy potential of the identified available land.

## 2. Materials and Methods

The methodology used in the present study has three overall steps: (i) crop selection, (ii) agroecological zoning, (iii) and energy yield estimation.

### 2.1. Crop Selection

A bibliographical review was performed to identify potential bioenergy crops for electricity production in Ecuador. The studies considered and addressed cultivars with the following desirable attributes: adaptability to latitudes near to zero degrees, a low moisture content, and a high energy yield GJ/ha per year [43,44,57–89]

The crops were divided into two groups: woody and non-woody biomass. The average yield for non-woody (Table 1) and woody (Table 2) biomass crops was 235 GJ/ha/year and 200.48 GJ/ha/year, respectively. The selected crops presented a higher energy yield per hectare than average.

**Table 1.** Non-woody biomass crops studied.

| Crop | Suitable to Be Produced Near the Equator (0° Latitude) | Yield t/ha/Year DM | Gross CV (DB) MJ kg | Gross Energy Potential GJ/ha/Year | Sources |
|---|---|---|---|---|---|
| Miscanthus (*Miscanthus* spp.) | X | 16.20 | 17.49 | 283.34 | [58] |
| Switchgrass (*Panicum virgatum* L.) | X | 10.20 | 18.00 | 183.60 | [58,60] |
| Reed canarygrass (*Phalaris arundinacea* L.) | | 5.50 | 17.83 | 98.06 | [59] |
| Virginia mallow (*Sida hermaphrodita*) | | 12.15 | 16.10 | 195.59 | [83] |
| Cardoon (*Cynara cardunculus*) | | 13.50 | 15.00 | 202.50 | [57] |
| Tall Wheatgrass (*Thino pyrumponticum*) | | 13.00 | 15.79 | 205.27 | [90] |
| Bamboo (*Bamboosa balcooa*) | X | 21.00 | 19.40 | 407.40 | [75,91] |
| Hemp (*Cannabis Sativa*) | X | 12.8 | 18.80 | 240.64 | [64,88,92] |
| Giant reed (*Arundo donax* L.) | X | 25.00 | 16.89 | 422.25 | [63,93] |
| Cup plant (*Silphium perfoliatum* L.) | | 6.70 | 17.19 | 115.17 | [84] |

**Table 2.** Woody biomass crops studied.

| Crop | Suitable to Be Produced Near the Equator (0° Latitude) | Yield t/ha/Year DM | Gross CV (DB) MJ/kg | Energy Potential GJ/ha/Year | Sources |
|---|---|---|---|---|---|
| Eucalyptus (*Eucalyptus globulus*) | x | 19.8 | 18.00 | 356.4 | [71,82,94] |
| Poplar hybrid | | 13 | 19.13 | 248.69 | [95–97] |
| Willow (*Salix* spp.) | | 18.6 | 19.33 | 359.5 | [98,99] |
| Pine (*Pinus patula*) | x | 6.5 | 19.30 | 125.45 | [65,100–102] |

The selected crops which present a higher energy potential per hectare than the average in their group are: for woody crops: eucalyptus and pine and from non-woody crops: miscanthus, giant reed, hemp, and bamboo.

### 2.1.1. Bamboo

Bamboo, *Bambusa balcooa*, family Bambusoideae, is a native crop from Northeastern India [78]. The dull green culms of this species are 12–23 m tall, with a 18–25 cm circumference, and they are widely scattered up to an altitude of about 600 m [103]. Because of the accelerated growth pattern with a short developmental phase, bamboo attains an essential position in cooperative agroforestry programs. Moreover, bamboo has been proven to be effective in controlling soil erosion [77].

### 2.1.2. Hemp

Hemp, *Cannabis sativa* L., family Cannabaceae, has been traditionally grown for its long bast fiber. Additionally, cannabinoids from hemp seeds have been used for medicinal, spiritual, and recreational purposes [104]. Hemp has been replaced as a manufacturing

material by cotton and synthetic fibers [105]. Hemp can produce high annual yields of biomass from 13 to 17.5 t/ha/year [64,88].

Furthermore, its advantages over other energy crops are relevant from an agricultural perspective, such as its good response to weeds and low fertilization requirements [106]. In Ecuador, [92] evaluated the yield of *Cannabis sativa* for biomass production and the average yield per harvest was 3.08 t of dry matter. The physiological maturity of the plant was achieved on the 86th day. Therefore, under normal conditions, four crops can be performed per year. In total, 12.8 t/ha can be harvested in a year. The yield data reported for studies conducted in Ecuador are in good agreement with international reports.

### 2.1.3. Eucalyptus

Eucalyptus, *Eucalyptus globulus*, family Myrtaceous, is a prime candidate for woody biomass plantations. It grows rapidly, presenting high dry matter yields per hectare per year, and can be planted in a wide range of agroecological conditions. The material harvested has 24% to 33% of dry weight. Eucalyptus trees are relatively deeply rooted and can obtain water and nutrients below the depths reached by most herbaceous perennial crops [82]. In Ecuador, commercial production system studies have presented a yield of 40 $m^3$/year or 19.8 t/ha [94].

### 2.1.4. Giant Reed

Giant reed, *Arundo donax*, family Poaceae, grows abundantly and presents with a high yield capacity [93]. Research carried out on this species has highlighted its high productive potential in several subtropical climates [93]. This plant has been studied as a potential carbon sink in European wetlands with promising results [107].

### 2.1.5. Pine

Pine, *Pinus patula*, family Pinaceae, is the most used species in forestry plantations due to its high cultivation yield that ranges from 12 to 22 $m^3$ per hectare per year [85]. Pine was introduced as a woody crop into Ecuador in 1925. Currently, pine accounts for 60,201.39 hectares [68,69,85]. The residues produced during pine processing provide the possibility of obtaining added-value products. The other advantages of pine in reforestation programs are related to its multiple uses and applications, its adaptability to degraded land, and its non-intensive agricultural practices [86]. Studies performed in Loja province (Ecuador) presented a yield of 12.19 $m^3$/ha/year or 6.2 t/ha/year of dry matter [102]. In addition, studies conducted in the Ecuadorian páramomeasured the carbon content of pine plantations at seven sites. The average results in terms of biomass yield were 6.5 t/ha/year. These data were used for the energy calculations [101] in this study. Consistency in the yield data was identified in studies developed in Ecuador.

### 2.1.6. Miscanthus

Miscanthus, *Miscanthus* spp., family Poaceae, is a native of Asia. The plant is a perennial grass with lignified stems; once the plant is established, Miscanthus presents stems larger than 3 m within a single growing season [108]. Miscanthus can maintain high photosynthetic rates followed by high biomass production [67]. It is suitable for various climates as perennial grass and has high water and nutrient use efficiencies. It is cultivable on marginal land without irrigation or heavy fertilization and is considered a leading energy crop [58].

### *2.2. Agroecological Zoning*

The agroecological zones (AEZ) methodology used in this part of the study was developed by the Food and Agriculture Organization of the United Nations (FAO), jointly with the International Institute for Applied Systems Analysis (IIASA) [4]. An agroecological zone is defined as homogenous and contiguous areas with similar soil, land, and climate characteristics. The methodology consists of the identification of areas with similar agroe-

cological characteristics taking into account biophysical and biochemical variables such as edaphic texture, depth, precipitation, stony, drainage, salinity, pH, fertility, slope, and climatic requirements such as temperature and the precipitation of a determined crop and the conditions given by an ecosystem and its evaluation about the aptitude for sustained use for some land use types (LUT). This information merges results and obtains limitations and potential areas to develop a determined crop [109].

Open-source software QGIS was used in this study [110]; the scale of the maps used is 1:25,000. The maps used edaphic factors (pH, salinity, slope, deep, texture, and drainage), land use, protected areas, isotherms, and isohyets to determine the optimal agricultural production areas in Ecuador within an agroecological zoning framework for each crop selected.

The basic criteria structure is described in Figure 1. The zoning performed considered the following criteria: (a) geopedological, including geomorphology, physical, and chemical properties of the soils, (b) climatic, including temperature and precipitation, (c) crop requirements information, including edaphic, physical, and chemical, and climatic requirements, and (d) production systems to be excluded: food security crops, ecological interest areas, and anthropic areas.

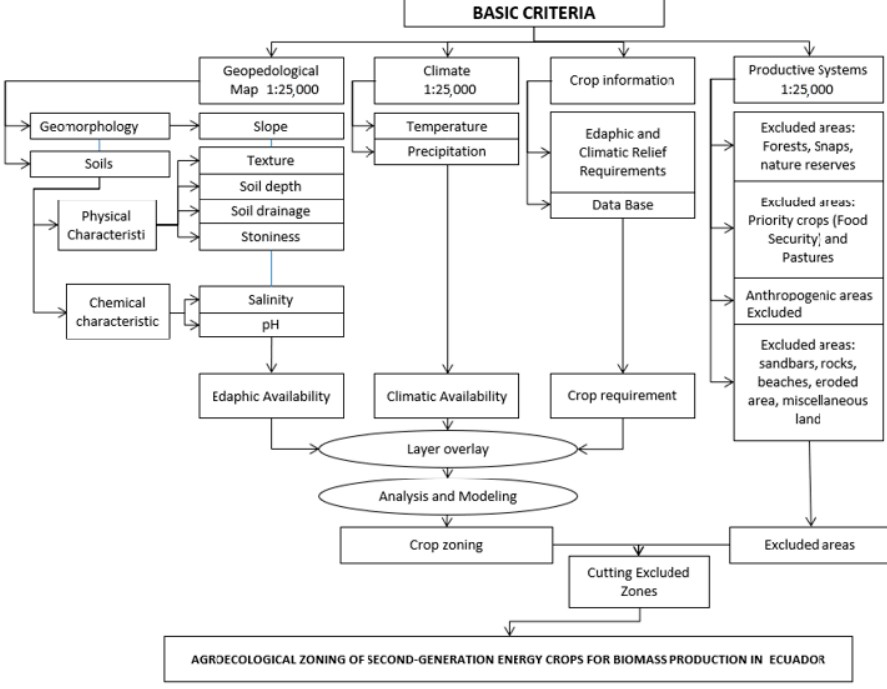

**Figure 1.** Agroecological zoning definition process for bioenergy crops in Ecuador.

The maps used were developed by the Ministry of Agriculture of Ecuador in the project: "Generation of geoinformation for the management of the territory and valuation of rural lands in the Guayas River basin" [111].

### 2.2.1. Agroclimatic Requirements of the Selected Crops

The biophysics and climatic variables for each selected crop were obtained from a bibliographical review; the sources are described in Table 3.

**Table 3.** Agroclimatic requirements of THE selected crops.

| Common Name | Giant Reed | Hemp | Miscanthus | Bamboo | Eucalyptus | Pine |
|---|---|---|---|---|---|---|
| Scientific Name | *Arundo donax* L. | *Cannabis sativa* | *Miscanthus* spp. | *Bambusa balcooa* | *Eucalyptus globulus* | *Pinus patula* |
| family | Poaceae | Cannabaceae | Poaceae | Poaceae | Myrtaceae | Pinaceae |
| Slope (%) | 0 to 5 | 2 to 25 | 0 to 25 | 0 to 40 | 2 to 25 | 2 to 25 |
| Soil Texture * | 1 to 13 | 2,3,4,5,8,9,10,11 | 3,4,5,7,8,9,10 | 8 to 12 | 3,4,5,7,8,9,10,11,12 | 2 to 11 |
| Effective soil Depth ** | 2 to 5 | 4 to 5 | 4 to 5 | 4 to 5 | 4 to 5 | 4 to 5 |
| Soil drainage | Excessive, good, moderate, poorly drained | Good, moderate | Good, moderate | Good, moderate | Good, moderate | Good, moderate |
| Soil stony | No stones, very few stones, few stones | No stones, very few stones, few stones | No stones, very few stones, few stones | No stones, very few stones, few stones | No stones, very few stones, few stones | No stones, very few stones, |
| Optimal pH | >5.5 to 8.5 | >5.5 to 7 | >5.5 to 7 | 5 to 6.5 | 5.5 to 7 | 5 to 6.5 |
| Optimal Temperature (°C) | 16 to 24 | 6 to 26 | 12 to 25 | 22 to 28 | 10.8 to 18 | 10 to 19 |
| Precipitation (mm) | 300–2000 | 600–1500 | 600 to 1400 | | 500 to 1500 | 700 to 1200 |
| Salinity (dS/m) | <2 to 16 | <2 | <2 | <2 | <2 | <2 |
| Sources | [43,112,113] | [64,104,105,114] | [61,66,108,115] | [75,76,80] | [72,73,116] | [85,117,118] |

\* Soil textures: sand (1), loamy sand (2) sandy loam (3), loamy (4), silty loam (5), silty (6), clay–sandy loam (7), loamy clay (8), silty clay loam (9) clay–sandy (10), clay–silty (11), clay (12), and heavy clay (13). \*\* Effective soil depth: very shallow soil (1), surface soil (2), shallow soil (3), moderately deep soil (4), and deep soil (5).

### 2.2.2. Excluded Systems

The proposed methodology excludes areas of ecological importance or those that are currently used for agricultural production, anthropic zones, and pastures of economic importance.

#### Agricultural Production Areas

Crops with economic and food sovereignty interests were identified based on their importance considering the following groups:

#### Export Crops

Crops for human consumption of export interests were identified using secondary information from the Central Bank of Ecuador [119].

#### Food Sovereignty Crops

The list of agricultural products commercialized on the Ecuadorian market was obtained and defined as food security; the information was obtained from the SIPA project [120]. The agricultural land use map was obtained from the Ministry of Agriculture of Ecuador (MAG). In this regard, the proposed methodology excluded areas dedicated to the agricultural production of 89 crops (Supplementary Materials Table S1).

#### Pastures

Pastures excluded from the study are defined in the land use classification as business and mercantile pastures, determining their economic importance for livestock production. Pastures defined as marginals have been considered within the zoning exercise because of their low profitability.

Ecologically Important Zones

Ecologically important zones inside the protected areas national system (PANE by its Spanish acronym).

The study aims to avoid interference with ecologically important areas; in this regard, the following land uses were excluded from the study: national reserve zones, protective forests, paramos, and zones declared by the State as reserves. The information was obtained from the natural heritage map of the national areas of Ecuador (PANE by its acronyms in Spanish) developed by the Ministry of Environment of Ecuador [121].

Ecologically Important Zones outside PANE

Disturbance or alteration of ecological systems is categorized as high, moderate, and low. In this study, there were also excluded zones of ecological importance that are not considered in PANE presenting moderate or low disturbance within the following categories of land use: forest, mangrove, paramo, shrub vegetation, and scrub zones considering their disturbance level were also excluded. This information was obtained through the national coverage and land use map developed by the Ministry of Agriculture (MAG by its acronyms in Spanish).

Anthropic Zones

This category considers all of the anthropic areas as infrastructure, roads, archaeological sites, landfills, and other anthropic zones. These areas were excluded from the study.

Areas of Water Sources

The excluded water sources are artificial water bodies, natural water bodies, residual water reservoirs, freshwater reservoirs, swamps, lava flow, glaciers, lakes/lagoons, snow and ice, and rivers.

Zones Classified as Miscellaneous

Many areas do not have soil or are very shallow; consequently, they support little vegetation, areas with no greater use; the rocky outcrop is an example. The names in the miscellaneous areas are used in the same way as names in soil taxonomy when identifying mapping units [122]. These excluded areas are rocky outcrops, flood areas, areas in the process of erosion, eroded areas, saline areas, sandbanks, and wasteland.

2.2.3. Map Overlay

Within a geographic information system (SIG) model, the union between the geopedological map and the isotherm and isohyet maps was performed within the required parameters for each crop. This step merged the information layers of the different maps to produce a fourth map with polygons containing geo-pedological and climate information. This process facilitates the creation of useful information for the development of the agroecological model. Finally, the areas mentioned in 3.2.10 were excluded. This process is illustrated in Figure 2 and was conducted for every selected crop.

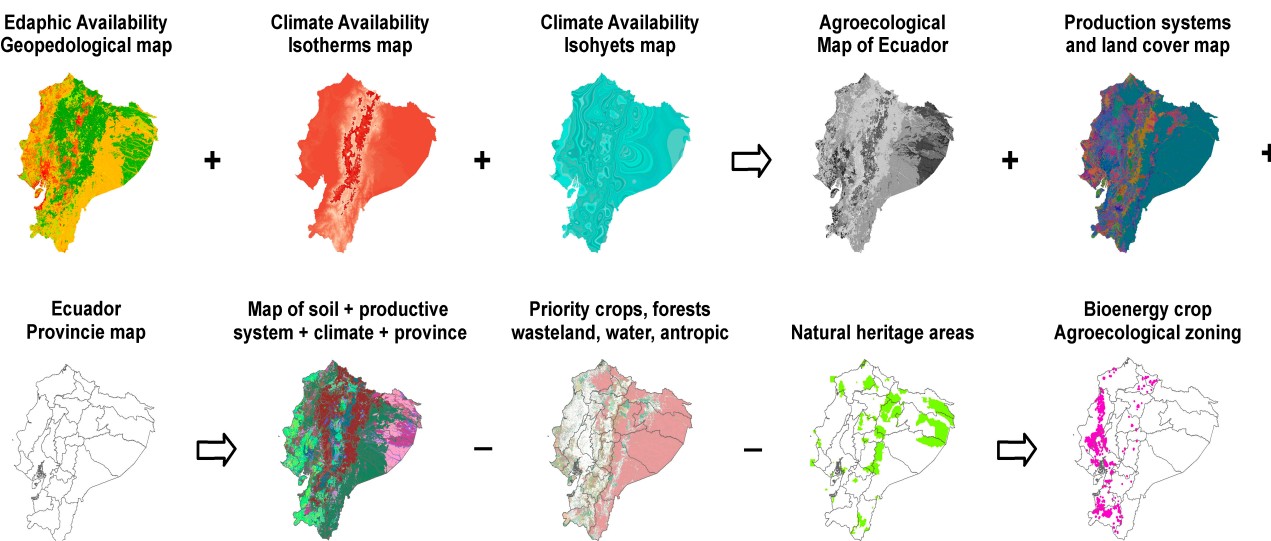

**Figure 2.** Agroecological zoning process to identify available land for bioenergy production in Ecuador, illustrative example for hemp.

### 2.3. Gross Energy Estimation

To determine the potential net energy yield of the estimated bioenergy feedstock production, a literature review was performed to determine the efficiency of electricity technologies available for biomass (Table 4). In this study, four main technological categories were identified for biomass plants, including reference values for size and efficiency: steam power plants, externally fired gas turbines, biomass-integrated gasification, and the combined cycle [123–126].

**Table 4.** Typical biomass energy generation technologies, configurations, and efficiencies.

| Configuration | Steam Power Plants (Backpressure Turbines) 20–25 MW | Steam Power Plants (Condensing Turbines) 5–50 MW | Externally Fired Gas Turbines 5–25 MW (Simple Cycle) | Externally Fired Gas Turbines 10–30 MW (Combined Cycle) | Biomass-Integrated Gasification 40–60 MW Simple Cycle | Combined Cycle 90–100 MW Combined Cycle |
|---|---|---|---|---|---|---|
| Lower limit Efficiency (%) | 10 | 22 | 25 | 30 | 21 | 35 |
| Upper limit Efficiency (%) | 20 | 28 | 30 | 40 | 25 | 40 |

Source: [123].

Equation (1) was used to calculate the potential energy production:

$$\text{Ney} = (\text{LCv} * \text{By}) * \text{Te} \tag{1}$$

Equation (1) is the crop net energy yield per hectare equation
Where:

Ney = net energy yield (MJ/ha)

LCv = lower caloric value of biomass (MJ/kg)

By = biomass yield on a dry basis (kg/ha)

Te = combined technology efficiency

Combined technology efficiency is a resulting energy efficiency figure that gathers all the referential conversion processes' efficiency percentages. Biomass pretreatment and chipping stages are neglected in this calculation.

Equation (2) was used to determine the potential energy yield per crop:

$$Eyc = Ney * Nha \tag{2}$$

Equation (2) is the crop net energy yield per hectare per year equation
Where:

$$Eye = \text{energy yield per crop per year (MJ/ha/year)}$$

$$Nha = \text{number of hectares (ha)}$$

In the case of overlapped areas, crops with the highest energy yield per hectare per year were prioritized over the rest, e.g, if the same area was shared by giant reed, Miscanthus, and Eucalyptus, giant reed was prioritized since it has the highest energy yield.

## 3. Results

Figure 3 shows the geographical location of the available areas to produce the selected bioenergy crops in Ecuador.

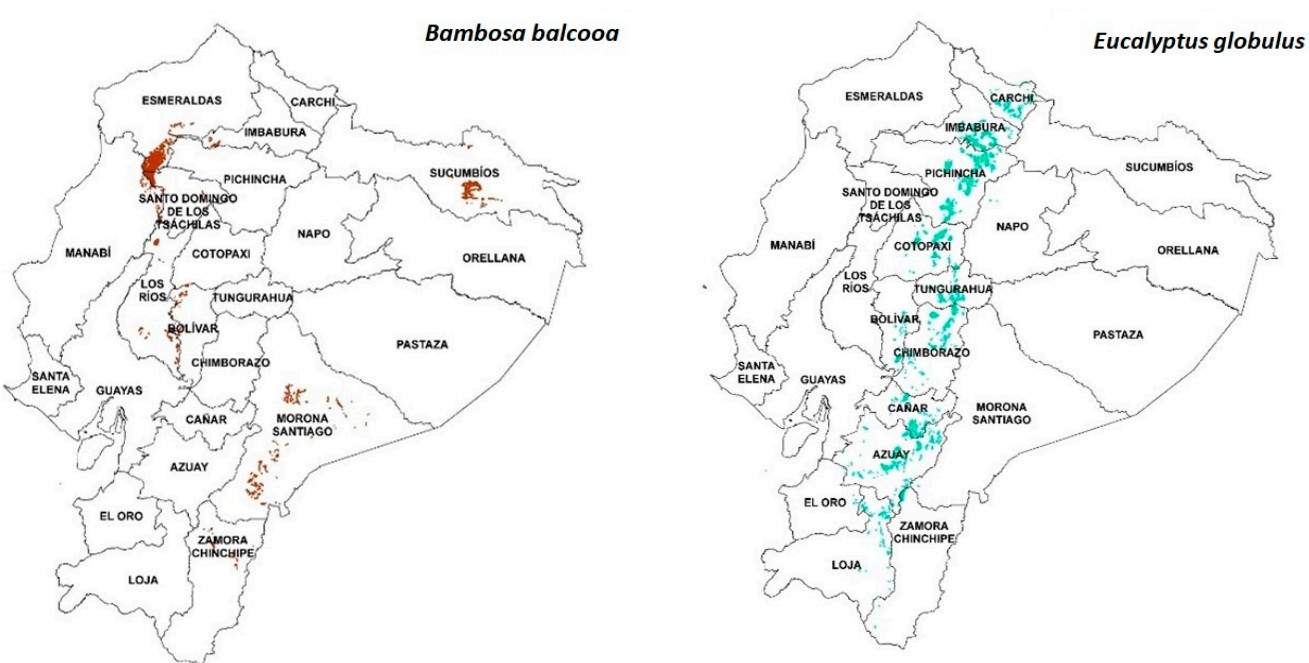

**Figure 3.** *Cont.*

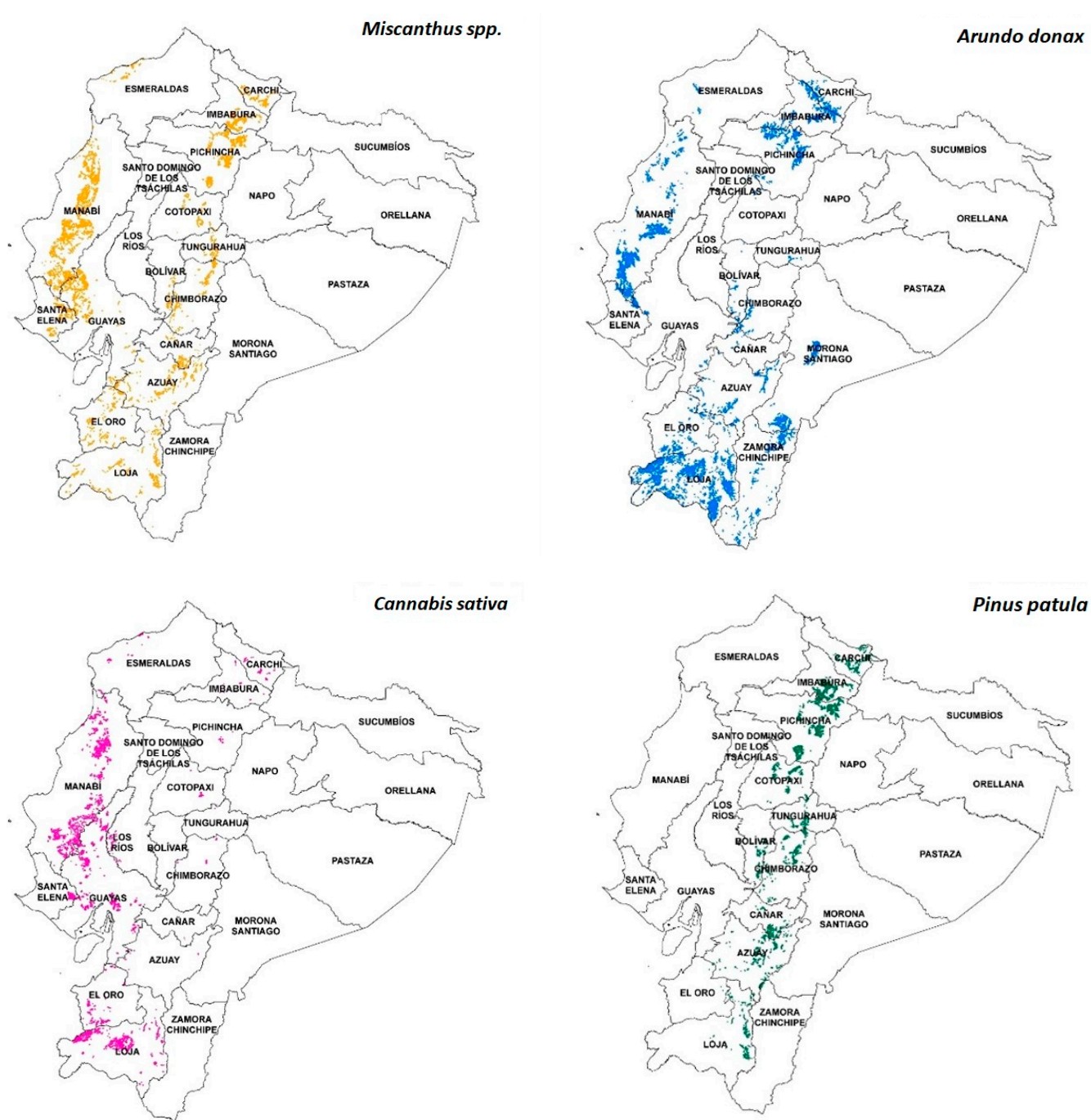

**Figure 3.** Available areas to produce bioenergy by selected crops in Ecuador.

### 3.1. Results per Crop

In terms of area, within the group of non-woody species, giant reed ranks first with 85,778 hectares available for its cultivation. In second place is Miscanthus, with 74,368 hectares. On the other hand, within the woody biomass group, Eucalyptus presents the largest area, 39,551 hectares, available for its cultivation. The results are shown in Figure 4.

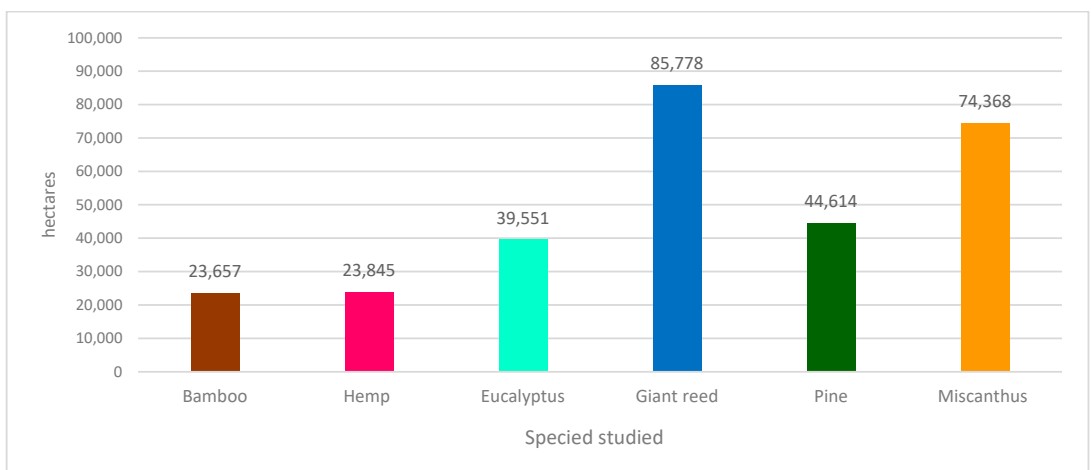

**Figure 4.** Total estimated area to produce bioenergy feedstock in Ecuador by crop.

### 3.1.1. Bamboo

According to the zoning results, there exists 23,635.15 ha available to produce *Bambusa balcooa* in Ecuador. Esmeraldas province presents the largest available area for Bamboo production, 10,460.87 ha, mainly in the Quinindé canton, followed by Morona Santiago province with 3968.90 ha (in the Morona, Limón Indanza, and San Juan Bosco cantons) followed by Sucumbíos province in the northeastern region of the country. These areas are characterized by warm, humid climates, and acid well-drained soils with frequent rain. Provinces located in the highlands present smaller optimal areas for Bamboo cultivation, mainly in warm microclimates.

### 3.1.2. Hemp

The study identified 23,844.59 ha available for *Cannabis sativa* production in Ecuador. Guayas province accounts for the larger area with 11,033.38 in the following cantons: Naranjal, Yagual, Yaguachi, Duran Balzar, Santa Lucia, Colimes, and Guayaquil. Manabí province presents the second largest area with 5449.94 ha, in the Pedernales, Flavio Alfaro, Chone, Santana, Olmedo Jipijapa, and 24 de Mayo y Pajan cantons. The province with the third largest area is Loja, mainly in the Puyango and Paltas cantons. The land use in these cantons is dominated by corn production, low to medium altered dry forests, and cultivated pastures. The vegetation in these provinces is characteristic of dry climates with high luminosity, non-saline soils, and low slopes. It is important to remark that small spots are identified within dry inter-Andean valleys among the Andean Mountain range. Furthermore, in the Amazonian region, no areas were identified due to its excessive precipitation and acid soils.

### 3.1.3. Eucalyptus

According to the agroecological zoning results, 39,550.53 ha in Ecuador are available to produce *Eucalyptus globulus*. Pichincha province has the largest available area with 10,804.46 ha in the Pichincha, Mejia, Pedro Moncayo, and Cayambe cantons. Cotopaxi, with 13,148.68 ha, is the second largest province with available land for producing the crop. The main cantons are Latacunga, Pujilí, Salcedo, Saquisilí, and Sigchos. The third province with the largest available land is Azuay, with 5297.75 ha in the Cuenca, Girón, Gualaceo, Paute, Chordeleg, El Pan, and Guachapala cantons. The vegetation in these areas is characteristic of cold and temperate climates, low precipitation, and wide soil ranges mainly present in the inter-Andean Mountain range.

### 3.1.4. Giant Reed

According to the agroecological zoning results, 85,777.66 ha in Ecuador are available to produce *Arundo donax* as a bioenergy feedstock in Ecuador. Loja province accounts for the

largest area with 25,296.27 ha in the following cantons: Puyango, Pindal, Paltas, Espindola, Loja, Macara, Olmedo, and Quilanga. The second largest area with 20,078.43 hectares is located in Manabí province in the following cantons: Junin, Jipijapa, and 24 de Mayo y Pajan. Due to its adaptability to a wide range of soil types, precipitation, drainage, and pH values, it has a presence in almost all zones with a prevalence in dry areas.

### 3.1.5. Pine

According to the agroecological zoning results, 44,614.41 hectares in Ecuador are available to produce *Pinus patula*. Pichincha province presents the largest available area, 13,344.45 ha. The main identified cantons with available areas are Distrito Metropolitano De Quito, Cayambe, Mejía, Pedro Moncayo. In Imbabura province, the second largest area was identified with 8843.38 ha, mainly in the following cantons: Ibarra, Antonio Ante, Cotacachi, and Otavalo. The third largest province with available land is Cotopaxi, with 5787.72 ha within the Latacunga, Pujilí, Salcedo, Saquisilí, and Sigchos cantons.

### 3.1.6. Miscanthus

According to the agroecological zoning results, there exists 74,367.89 ha in Ecuador available to produce *Miscanthus* spp. The largest available area of 16,789 ha is in Manabí province in the Portoviejo, Bolivar, Junín, Chone, Pajan, Tosagua, Santana, and 24 de Mayo cantons. The second province with the most available area to produce Miscanthus is Guayas, with 13,148.68 ha, mainly in the Pedro Carbo and Isidro Ayora cantons. These crops and vegetation respond to low precipitation, moderately acidic pH values, and hot–dry climates.

### 3.2. Energy Yield Estimation per Crop

Four main technological categories of bioenergy plants were identified, including reference values for size and efficiency: steam power plants, externally fired gas turbines, biomass integrated gasification, and combined cycle power plants. Using the information mentioned above, the net potential energy yield per technology with different capacities and efficiencies was estimated. The results are shown in Table 5. Giant reed ranks first with a potential net energy yield of 4,024,401.74 MWh per year using combined-cycle and gas turbines technologies.

**Table 5.** Potential net energy yield estimation by technology per crop, in MWh/year.

| Technology | Gross Energy Potential MWh/Year | Bamboo 2,677,144 | Hemp 1,593,878 | Eucalyptus 3,915,503 | Giant Reed 10,061,004 | Pine 1,554,688 | Miscanthus 5,853,166 | Total 25,655,384 |
|---|---|---|---|---|---|---|---|---|
| Steam power plants (backpressure turbines) 20–25 MW | Lower limit 10% (efficiency) | 267,714 | 159,388 | 391,550 | 1,006,100 | 155,469 | 585,317 | 2,565,538 |
| | Upper limit 20% (efficiency) | 535,429 | 318,776 | 783,101 | 2,012,201 | 310,938 | 1,170,633 | 5,131,077 |
| Steam power plants (condensing turbines) 5–50 MW | Lower limit 22% (efficiency) | 588,972 | 350,653 | 861,411 | 2,213,421 | 342,031 | 1,287,697 | 5,644,184 |
| | Upper limit 28% (efficiency) | 749,600 | 446,286 | 1,096,341 | 2,817,081 | 435,313 | 1,638,886 | 7,183,507 |
| Externally fired gas turbines 5–25 MW (simple cycle) | Lower limit 25% (efficiency) | 669,286 | 398,470 | 978,876 | 2,515,251 | 388,672 | 1,463,291 | 6,413,846 |
| | Upper limit 30% (efficiency) | 803,143 | 478,163 | 1,174,651 | 3,018,301 | 466,406 | 1,755,950 | 7,696,615 |
| Externally fired gas turbines 10–30 MW (combined cycle) | Lower limit 30% (efficiency) | 803,143 | 478,163 | 1,174,651 | 3,018,301 | 466,406 | 1,755,950 | 7,696,615 |
| | Upper limit 40% (efficiency) | 1,070,858 | 637,551 | 1,566,201 | 4,024,402 | 621,875 | 2,341,266 | 10,262,153 |
| Biomass integrated gasification 40–60 MW simple cycle | Lower limit 21% (efficiency) | 562,200 | 334,714 | 822,256 | 2,112,811 | 326,485 | 1,229,165 | 5,387,631 |
| | Upper limit 25% (efficiency) | 669,286 | 398,470 | 978,876 | 2,515,251 | 388,672 | 1,463,291 | 6,413,846 |
| Combined cycle 90–100 MW combined cycle | Lower limit 35% (efficiency) | 937,001 | 557,857 | 1,370,426 | 3,521,352 | 544,141 | 2,048,608 | 8,979,384 |
| | Upper limit 40% (efficiency) | 1,070,858 | 637,551 | 1,566,201 | 4,024,402 | 621,875 | 2,341,266 | 10,262,153 |

### 3.3. Energy Yield Estimation per Province

To determine the energy potential per province, a crop map overlay was developed to determine the intersection areas where two or more crops share the same area. Figure 5 illustrates the intersection of the crops identified.

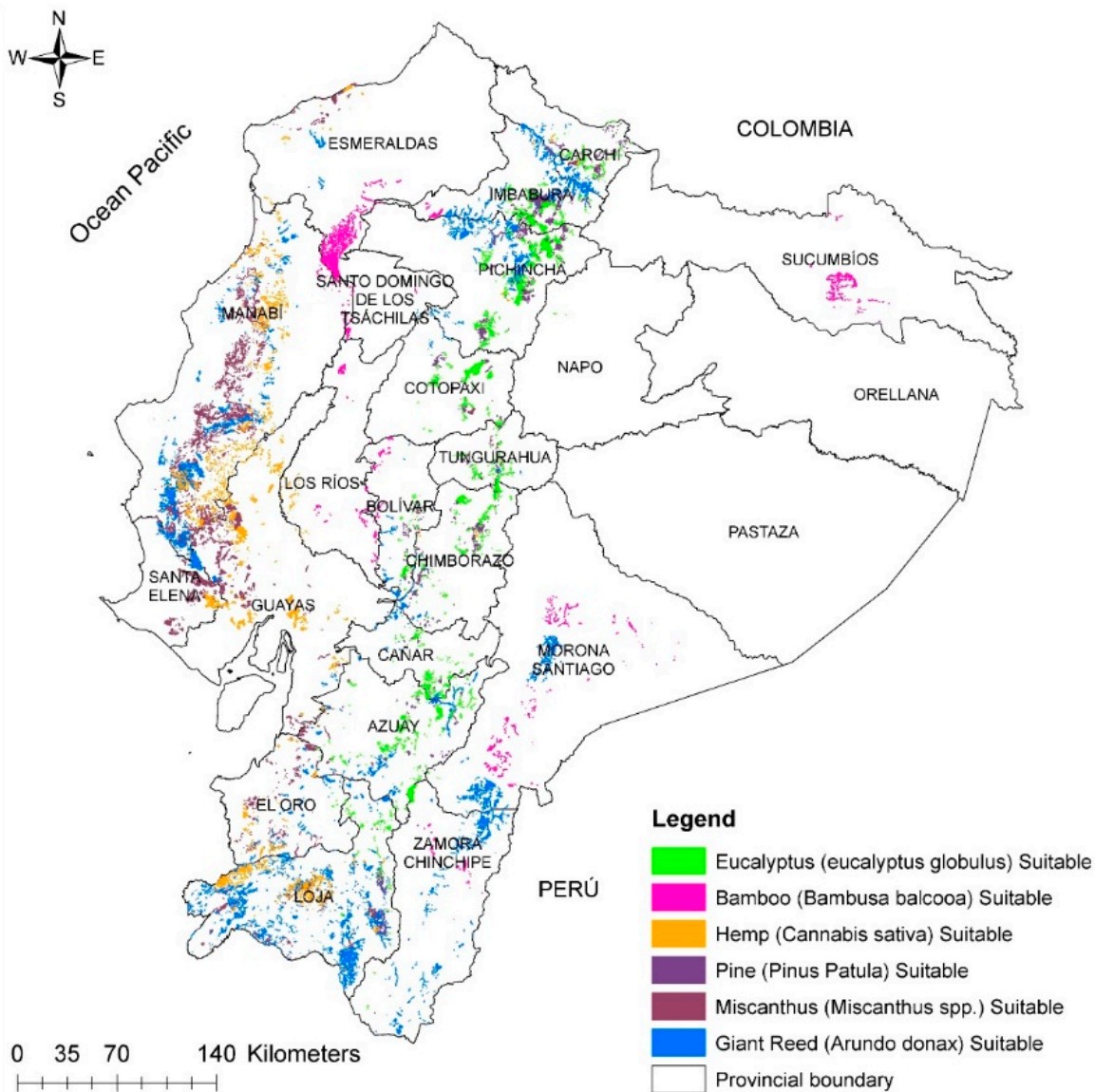

**Figure 5.** Intersection of resulting areas for the selected biomass crops.

Once the intersected areas were identified, the crops with the highest energy yield per hectare per year were prioritized over the rest if sharing the same area, e.g., if the area was shared by giant reed, Miscanthus, and Eucalyptus; giant reed was prioritized since it has the highest energy yield according to Tables 4 and 5. Finally, in this part of the study, the resulting number of hectares was multiplied by the potential gross energy yield per crop per year, as shown in Figure 6. The total net energy yield per province and technology is shown in Supplementary Materials in Table S2.

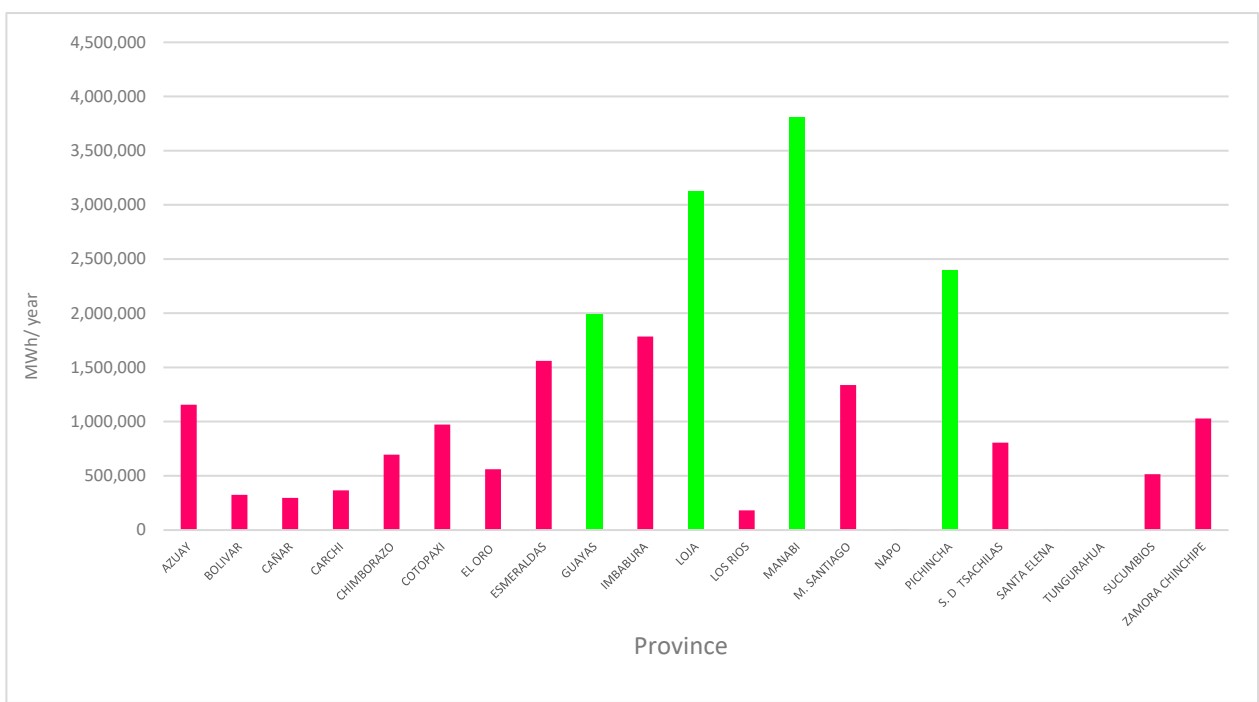

**Figure 6.** Gross bioenergy potential MWh/year in Ecuador per province.

## 4. Discussion

### *4.1. Ecuadorian Context*

Ecuador must consider the increase in the electricity demand, hydroelectricity deployment constraints, and the carbon intensity of thermal electricity generation technologies to achieve its NDC goals. Ecuador must implement programmatic approaches to promote medium- and long-term baseload electricity resources such as biomass. Furthermore, to achieve deep decarbonization in the country, it is necessary to foster strategic climate change mitigation scenarios in which bioenergy and reforestation could play a central role as indicated by [18].

According to the results of the Bioenergy Atlas of Ecuador, the gross energy potential of the by-products of the main agricultural crops in the country is 199,057.23 TJ per year [28]. On the other hand, the results of the present study show a gross energy potential of 82,466.92 TJ per year. It must be noted that although the energy potential of agricultural waste doubles the one of dedicated energy crops, there still exists logistical and regulatory constraints that have limited its deployment.

The agroecological zoning performed for Ecuador resulted in the potential energy production of 8603 GWh per year. This number represents 29% of the electricity demand in 2019, and 8.8%, 4.9%, and 5.5% of the three projected electricity generation scenarios for the year 2050 developed by IIGE [127].

### *4.2. Critical Factors for Zoning Studies*

Ref. [128] recommended critical factors to be considered when addressing biomass potentials. The following section contrasts some of these factors with the methodology applied in the presented study and the bibliographical review.

#### 4.2.1. Land Management

Large-scale bioenergy production will transform land use patterns, which pose challenges to land management. Extensive land use associated with bioenergy production can modify regional and global climatic processes and should be carefully considered by policymakers and planners. It is equally important to consider how the changes in land

use to accommodate bioenergy crops affect the mix of goods and services as food, fiber, fuel, freshwater, recreation, among others that are provided by land [129].

In this regard, land management planning must consider all possible interactions between the mentioned components to avoid major land management related conflicts. As mentioned above, before the development of bioenergy projects, it is recommended to perform a full assessment of different land management options considering food security [2] and indirect land-use change [9]. In addition, other ecological, economic, and social land use change related impacts of the potential bioenergy production also deserve additional assessment.

Improvements in agricultural and forestry management and technologies were factors not considered in this study. The increment of yield per agricultural unit due to efficient farming practices could reduce the amount of land required for food production; this could imply an increment in the availability of land for bioenergy production.

Choice of energy crops: variables to consider within these criteria are the impact of energy conversion efficiency and higher energy yields between woody and herbaceous plants when compared to grains and oils, avoidance of food-fuel conflicts, as well as inputs requirements. Although most of these criteria were addressed in the present study, it is recommended to analyze further aspects as the comparison of woody and herbaceous with grains and oils and inputs requirements. The bioenergy potential of non-edible vs. edible crops is a critical issue to be addressed before decision making. One of the limitations of this study is that production yields correspond to literature values. Therein, agricultural experimentation and yield evaluation under Ecuadorian conditions in the recommended areas is crucial before starting commercial production of any feedstock presented in this study.

Economic parameters: economic factors as market mechanisms and value chain associated cost of biomass production were not included in the evaluation. The present study aimed at exploring the potential for bioenergy dedicated crops from an agronomical perspective. Nevertheless, these important variables should be considered in cost–benefit decision-making analysis when this information is available for Ecuador. Moreover, it is important to integrate holistic approaches such as life cycle sustainability assessment (LCSA) to evaluate the environmental, social, and economic impacts to support decision-making processes towards bioenergy feedstock selection.

### 4.2.2. Food Security

Large-scale cultivation of dedicated biomass is likely to affect global food prices and water scarcity. Biomass from cellulosic bioenergy crops is expected to play a substantial role in future energy scenarios to achieve decarbonization goals. Nevertheless, the worldwide potential of bioenergy is limited because all land is multi-functional, and land is also needed for food, feed, timber, fiber production, and nature conservation and climate protection [9].

Hasegawa et al. (2020) [130], suggest that extensive bioenergy production, if not implemented properly, may cause food prices raise and increase the number of people at risk of hunger in many areas of the world. For example, an increase in global bioenergy demand from 200 to 300 EJ could cause a $-11\%$ to $+40\%$ change in food crop prices and decreases food consumption from $-45$ to $-2$ kcal person$-1$ day$-1$, leading to an additional 0 to 25 million people at risk of hunger compared with a no bioenergy demand scenario.

Therefore, multidisciplinary approaches, as the Bioenergy and Food Security (BEFS) Appraisal, can provide the first level of information for a better decision-making process based on biomass available for bioenergy production and specific bioenergy supply chains. The model evaluates the bioenergy production potential by quantifying the feedstock available, identifying income and employment opportunities, and energy access options [56].

The methodological approach used in this study excluded productive zones dedicated to the main agricultural products from food safety and economic perspectives in the country.

### 4.2.3. Indirect Land Use Change

Indirect land use change (iLUC) refers to the changes in land use that occur elsewhere because of a bioenergy production project. As an illustrative example: displaced food producers may re-establish their operations elsewhere by converting natural ecosystems to agricultural land, or because of macroeconomic factors, the agriculture area may expand to compensate for the losses in food/feed/fiber production caused by a bioenergy project [131].

Searchinger (2013) [132] argued that bioenergy production is likely to increase GHG emissions relative to the fossil fuels they were intended to replace because of emissions attributable to 'indirect land use change' (ILUC). Biofuel production can reduce the supply of feedstocks available for feed or food, leading to higher prices and land pressure. Farmers respond to higher prices by increasing the feedstock supply, which expands the footprint of farming into natural land [133]. This issue must be addressed through careful land use planning, considering all of the possible interactions and consequences of bioenergy cropping. Models based on supply and demand, as developed by [134], can be applied to foreseen circumstances and avoid undesired iLUC-related impacts.

The present study utilizes a top-down approach that evaluates the current land use, excludes human or ecologically important areas, and identifies available land for specific bioenergy crops according to its edaphoclimatic requirements. Nevertheless, the identification of degraded land is not an objective of this study.

### 4.2.4. Ecological Risks

Environmental issues need to be addressed when dealing with non-native species for bioenergy ends, including invasiveness potential, fire risk, water use, and sustainability [135].

Due to extensive bioenergy monocultures, ecological impacts can be linked to biodiversity loss [136], reduced water yield, and increased nutrient load, which may contribute to hypoxic conditions and eutrophication. To mitigate the potential adverse environmental impacts, best management practices (BMPs) such as conservation tillage, and filter strips, among others, are a set of methods to be considered as part of sustainable bioenergy production [40].

Environmental impacts related to water use are very important to stress in crops such as Eucalyptus, whose root system has evolved to reach water sources such as underground aquifers, which can reduce ecological water flows. Nevertheless, physiological studies in several countries have shown that Eucalyptus has a similar water use efficiency (WUE) to other tree species. Water consumption at the stand level depends upon water availability, vapor pressure deficit, and WUE; water availability, therefore, is a major determinant of productivity. Actual water use by Eucalyptus in a watershed depends on many factors, including the areal extent, size, spatial distribution, productivity, and age-class distribution of planted stands. Eucalyptus has potentially higher water use and water use efficiency than pasture, pine plantations, and native forests, but water use is much lower in Eucalyptus plantings than in irrigated crops. Studies in other countries suggest that the effects of Eucalyptus plantations on stream flow may be most apparent in drier regions where precipitation is approximately equal to evapotranspiration. Water consumption by Eucalyptus plantations will be higher in terms of the percentage of water supply in drier regions, but absolute water consumption will be higher in the wetter region [137].

Thus, quantifying the long-term effects of land management changes in bioenergy projects is critical for developing adaptive strategies for water resource and ecosystem sustainability in general terms. Project developers must consider approaches that reduce these types of threats as keeping plantations away from watercourses and maintaining clear firebreaks. Interspersing potential invasive species with others can reduce ecological threats. By planting sterile genotypes, short rotations reduce the total flower production potential of individual trees and stands and reduce the total number of heavy seed production years [135].

The present study assumes no irrigation,. However, irrigation could increment available zones for dedicated bioenergy crops; water distribution among food and energy crops must be carefully studied. In addition, current and future water availability should be considered in further research. Moreover, the water footprint (WF) for biomass production is a recurring issue [128]. For most crops, the WF of bioelectricity is about a factor of two smaller than bioethanol or biodiesel [137].

It must be highlighted that ecologically important zones were excluded from the zoning performed. Nonetheless, aspects as the future expansion of natural protected areas, long term climate change impacts, shifts in vegetation zones, and bioenergy-induced eutrophication and acidification, among other facts, must be considered in future studies. The increment on protected areas could reduce the available surface for bioenergy production. The present study identified advance ecologically important zones not considered currently in the national protected areas system.

### 4.2.5. Climate Change and Greenhouse Gas Emissions

The present work does not determine the global warming potential associated with bioenergy production from the studied crops. It is recommended for this issue to be addressed in future studies. Moreover, the projection of future climate change scenarios would change the future availability of the resulting areas identified in this study.

### 4.3. *Limitations and Recommendations for Future Work*

One of the main limitations of this work is the preselection of species from the literature review in different latitudes due to the lack of commercial production experiences in Ecuador in some cases, such as for Poplar spp. It must be noted that its distribution is predominant in temperate regions above 28° latitudinal limits of northern and southern hemispheres [138]. There is not a lot of information available on commercial growing of the species at low latitudes and away from their natural distribution range. As exotic plantation species, the genera Populus and Salix yield exceedingly good results in South America, especially in Argentina (latitude 34°) and Chile (between latitude 32° and 36° S) [139].

There are some reported punctual experiences of production in lower latitudes (18° S) in Zimbabwe of *P. alba*/*P. x canescens* and the natural occurrence of *Populus ilicifolia* in Kenya; nonetheless information related to its production is hardly available, moreover, the latter is included in the Red List of threatened species of the International Union for Conservation of Nature (IUCN) due to its habitat lost [140]. Nevertheless, the species mentioned are interesting prospects to further adaptation studies to Ecuadorian conditions.

The same limitation applies for the species of bamboo studied *Bambusa balcooa*, taking into consideration that there exists a total number of 1400 bamboo species distributed worldwide [139], a more detailed study of prospects adaptable to Ecuador must be conducted.

The zoning developed is based on the current land use; nevertheless, future market demand can modify the land use and compete with potential bioenergy crops. These possible interactions must be considered in future works.

The current study utilizes geographical information produced by governmental entities; the variables studied and the scales can change over time; in this regard, it is important to recommend using updated GIS material when commercial production decisions are considered. The only bioenergy sources currently considered in the prospective energy scenarios for Ecuador are bagasse and firewood [127]. It is recommended to consider integrating the energy potential of dedicated bioenergy crops, as described in this study, when energy prospective scenarios are developed for Ecuador.

It is crucial to address the fact that although exhaustive bibliographical research regarding the production of the identified crops in Ecuador was performed. It is understood that yields will not be uniform across the range of soils and the different agroecological zones identified. Therefore, it is highly recommended that future studies address this issue

with strategically placed experimental designed trails for evaluating yield and the overall adaptation of the crops within each zone identified.

Finally, it is highly recommended that future studies consider other nontraditional, promising bioenergy crops adaptable to Ecuadorian conditions.

## 5. Conclusions

The production of bioenergy from dedicated biomass crops is a valid alternative to support the decarbonization of electricity matrices in line with global accords. Nevertheless, conflicts related to food safety and potential environmental impacts are a persistent issue extensively addressed in the literature.

This work proposes a new conceptual framework for bioenergy production taking into consideration the avoidance of food production systems and zones of ecological and atrophic importance. The present study identified dedicated bioenergy crops that could be produced in Ecuador. Using GIS, the availability of land with suitable agroecological conditions for the identified crops was addressed within a non-competitive scheme. A Potential energy production of 8603 GWh per year, which represents 29% of the projected electricity demand in 2019, was determined. Nevertheless, future decarbonization scenarios project a need of 1244 GWh from biomass resources by the year 2050 [140]. Therefore, further research on additional resources must be considered to address the projected demand.

The potential ecological, food safety, and social-related impacts of bioenergy-dedicated crops are issues that must be addressed carefully when conducting landscape development planning. Indirect land use change and land management impacts need to be considered and analyzed deeply on a case-by-case basis. On the other hand, economic performance indicators, such as opportunity costs evaluations, are components that are considered in the literature and can be integrated into zoning planning. Moreover, the thresholds of the defined productive zones can be modified by the climate conditions, market conditions, policy incentives, or disincentives.

The results of this study can serve as a basis for policy decision-making as well as private endeavors. Nevertheless, before the implementation of a bioenergy project considering non-traditional or non-native crops, future research, including the implementation of in situ trials, is needed. The participation of local research institutes and private stakeholders is a crucial success factor.

The complementarity of bioenergy with carbon capture and storage (BECCS) should be studied deeply in Ecuador, considering the opportunities that can arise from the potential improvement of carbon markets. Finally, considering the results obtained in the present study, bioenergy-dedicated crops are a valid option to be considered and integrated into the energy planning of Ecuador.

**Supplementary Materials:** The following supporting information can be downloaded at: https://www.mdpi.com/article/10.3390/agriculture13010186/s1, Table S1: Excluded crops considered for food safety; Table S2: Energy yield per technology by province MWh/year.

**Author Contributions:** Conceptualization, C.R.P., L.M.N.-G. and A.D.R.; methodology, C.R.P., A.C.-G. and A.D.R.; investigation, C.R.P., A.D.R., L.M.N.-G., A.C.-G. and D.G.; writing—original draft preparation, C.R.P.; writing—review and editing, A.D.R., A.C.-G. and L.M.N.-G.; visualization, C.R.P.; supervision, A.C.-G. and A.D.R.; project administration, L.M.N.-G. and A.D.R. All authors have read and agreed to the published version of the manuscript.

**Funding:** This research received no external funding.

**Institutional Review Board Statement:** Not applicable.

**Data Availability Statement:** No new data were created or analyzed in this study. Data sharing is not applicable to this article.

**Acknowledgments:** This research work is the part of Ph.D. Thesis of Christian R. Parra at the University of Valladolid (Spain). Christian R. Parra would like to thank to the SIGTIERRAS program of the Ministry of Agriculture, Livestock, Aquaculture and Fisheries of Ecuador.

**Conflicts of Interest:** The authors declare no conflict of interest. The funders had no role in the design of the study; in the collection, analysis, or interpretation of the data; in the writing of the manuscript, or in the decision to publish the results.

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
