# Peer review of "Prospects for Bioenergy Development Potential from Dedicated Energy Crops in Ecuador: An Agroecological Zoning Study"

_agriculture, doi:10.3390/agriculture13010186_

Round 1

Reviewer 1 Report

The work of Parra et al. presents the overview of the Ecuador biomass valorisation for bioenergy. This work demonstrates some potential, however authors should improve their work before this work can be accepted for publication.

For example, the introduction section should be rewritten to include the fact that bioenergy is just a part of the biorefinery concept [1] and circular economy [2]. In addition, authors should include in their work the aspects related to the use of presented crops for the alternative applications. To what extent the use of presented crops affects the current other use? What are the alternatives for the formation of additional demand on the market.

After these corrections, this work can be accepted for publication.

[1]         P. Manzanares, The role of biorefinering research in the development of a modern bioeconomy, Acta Innov. 37 (2020) 47–56. https://doi.org/10.32933/actainnovations.37.4.

[2]         P. Morone, G. Yilan, A paradigm shift in sustainability: from lines to circles, Acta Innov. 36 (2020) 5–16. https://doi.org/10.32933/actainnovations.36.1.

Author Response

Please, see the attached document

Reviewer 2 Report

This manuscript deals with a relevant issue: identification and development of alternative feedstock for bioenergy production.

This study identify potential sustainable bioenergy resources for continental Ecuador using agroecological zoning methodologies for non-food crops. "The research identify available land with suitable agroecological conditions to produce dedicated bioenergy crops within a non-competitive scheme with food and natural ecosystems and quantify the gross bioenergy potential of the identified available land. " This approach is innovative.

The research issue, objectives and method are clear and coherently stated.

The research results are clear and soundly presented and discussed.

The conclusions are pertinent and supported by the reseach results.

Nevertheless, some points are worth revisiting:

1 - The abstract is generic in nature. The main research results and policy and managerial implications could be a little more detailed (to explore the content from section 4.2 - Critical factors for zoning studies and );

2 - The Figures numbers are repeated and must be revisited.

3 - Table S2 (supplementary material) could be moved to the main text(manuscript).

Author Response

Please, see the attached document

Reviewer 3 Report

The authors of the paper "Prospects for bioenergy development potential from dedicated energy crops in Ecuador: An agroecological zoning study" available to me have written a very interesting and effective article. Unfortunately, the documents and literature examples used are so incomplete or incorrect that I cannot recommend this article for publication in its present form.

The authors completely fail to explain

1.      why they concentrate only on a single bamboo species (Bamboo balcooa) in this study given the sheer inexhaustible abundance of different bamboo species,

2.       why do they rate the possible yields of poplars so low and

3.       why do they write that with the abundance of different poplar varieties, divided into 6 sections (according to Eckenwalder (1996), they claim that poplars are not possible for cultivation in subtropical regions. I quote a short excerpt from Isebrands, J.G. and Richardson, J., which was written in 2014: “The genus Populus is not found in the southern hemisphere, with the exception of populations of P. ilicifolia native to Kenya and Tanzania……As exotic plantation species, however, Populus and Salix are yielding exceedingly good results in South America, especially in Argentina and Chile. This is significant, because South American nations are known globally as leaders in the production, marketing and science of exotic plantation forestry. The plantation production of Monterey pine (Pinus radiata) and various eucalyptus species (Eucalyptus spp.) throughout Argentina, Brazil, Chile and Uruguay has made significant contributions to the global wood products industry over the past 30 years. Populus and Salix plantations have not approached the economic importance of these two genera, but may do so some day. Historically, poplar has been grown in South America for veneer logs for the matchstick industry. This market remains a strong outlet, although poplar, along with willow, is now being grown increasingly for the pulp and paper and medium-density fibreboard industries. Initiatives are also under way to grow selections from both genera as feedstock for the renewable energy industry.

I also find it very confusing when the authors speak of P. tremula in Table 2, but P. alba is mentioned in the literature which the authors refer [94]. Apart from that, a single source of literature for a decision with such far-reaching consequences is, in my opinion, more than poor.

I would urge the authors to be more open about the variety of possibilities when describing prospects and not to get stuck in the tradition of the last 150 years.

Apart from this fundamental and serious error, I noticed a few technical weaknesses that the authors should take into account in a possible new submission:

L. 141 and 142: The numbering of the tables is exactly the other way around.

When listing the favorite plants in L. 154 and 155, herbaceous plants and trees are suddenly listed in a jumble, although they were previously neatly separated: why?

The scientific names of the plants must ALWAYS be written in italics, with the species name in lowercase.

The authors should follow the philosophy that the table headings and figure captions should be self-explanatory on their own. The way it was done in this paper is not only far too short, it is simply incomplete.

Figure 1 on page 11 is actually Figure 3.

All graphs must have a full axis label: I miss the x-axis label in Figures 4 and 6.

Literature 77 is misquoted: Benton, A. is merely a chapter in a book, which must be given with the editors.

References:

Eckenwalder, J.E. (1996): Systematics and evolution of Populus. In: R.F. Stettler et al.: Biology of Populus and its implications for management and conservation. 1996, S. 7–32.

Isebrands, J.G. and Richardson, J. (2014): Poplars and willows: trees for society and the environment ISBN 978-1-78064-108-9 (hbk) -- ISBN 978-9251071854 (co publisher FAO)

Author Response

Please, see the attached document

Round 2

Reviewer 1 Report

The authors included the suggested changes and as such this work can be accepted for publication in the current form.

Author Response

We want to thank the reviewer for his kind words. He decided to accept the article in its current form

Reviewer 3 Report

The authors of the paper "Prospects for bioenergy development potential from dedicated energy crops in Ecuador: An agroecological zoning study" have revised their paper according to my instructions and have already gone "in the right direction".

Nevertheless, I think that there are still some things to be worked out before the paper can be published:

·         I find it only logical and honest when the authors clearly state the limits of their work both in the headline and in the abstract: these are more or less traditional cultures that are respected!

·         Unfortunately, the authors still underestimate the productivity of hybrid poplars in Table 2. Please see e.g. the paper by Zalesny et al. (2016), there reports of harvest quantities of up to 13 dry Mg ha−1 year−1 in the southern states of the USA and thus quite comparable climatic conditions.

·         I would also recommend placing the results on a broader basis of literature research. In recent years, an incredible number of papers have been published on the potential yield of poplar and willow hybrids.

·         Unfortunately, the authors have not succeeded in transcribing all scientific names of plants in italics and writing the species name in lower case in the literature.

References

Zalesny. R.S., Stanturf, J.A., Gardiner, E.S., Perdue, J.H., Young, T.M., Coyle, D.R., Headlee, W.L., Bañuelos, G.S., Hass, A. (2016): Ecosystem Services of Woody Crop Production Systems. Bioenerg. Res. (2016) 9:465–491, DOI 10.1007/s12155-016-9737-z
